# Position: If open source is to win, it must go public

Joshua Tan [1 2]  Nicholas Vincent [3 1]  Katherine Elkins [4]  Magnus Sahlgren [5]  Joseph Low [1 6]  David Pham [3 1]
Sampo Pyysalo [7]  Jenia Jitsev [8 9]

## Abstract

Open source projects have made incredible progress in producing widely usable machine learning models and systems, but open source alone will face challenges in fully democratizing access to AI. Unlike previous generations of open source software, open source and open weight AI models require substantial resources to activate and maintain—e.g., data and compute for pre-training, post-training, and deployment—which only a few actors can currently provide. This position paper argues that open source AI must be complemented by public AI: infrastructure and institutions that ensure models are accessible, sustainable, and governed in the public interest. To achieve the full promise of AI models as prosocial public goods, we need to build public infrastructure to power and deliver open source software and models.

## 1. Introduction

Open source, and the ethos of openness, has long served as a counterweight to concentrated control in computing. From Linux to Kubernetes, open collaboration has enabled researchers, companies, and the public at large to build on shared and trustworthy infrastructure (Raymond, 1999; Eghbal, 2016). But open source has always straddled a line between the emancipatory ideals of the Free Software movement and the strategic goals of firms (Weber, 2005; Kelty, 2008). What appears as a spontaneous gift economy is often scaffolded by sponsorships, employment arrangements, and various private (and public) subsidies. This compromise between community and commerce has proven to be remarkably productive in many software categories such as

cloud computing, programming languages, and operating systems. But it is breaking down for the largest foundation models in AI. Such large language models (LLMs) are incredibly expensive to train, raising questions about the longevity of open models (Maslej et al., 2025; Choksi et al., 2025). Once trained, open source weights alone are inert; without inference, fine-tuning, localization, tooling, and interfaces, they remain unusable to all but a small elite with the capital, compute, and engineering to deploy them (Bommasani et al., 2024; HAI, 2024). And even if deployed, the decentralized nature of open source deployments means that important RLHF and query data can become stranded across many silos. As modern AI ecosystems mature, so too must our expectations about what open source practices can–and cannot–deliver for researchers, for firms, and for the public.

In this paper, we argue that **without structural intervention from public institutions, current open source efforts in AI will not democratize access to AI nor provision public goods** (in the technical sense of non-rivalrous and non-excludable goods) as comparable open source efforts have done in other categories. This will hurt the machine learning research community. It will hurt startups. It will also undermine the strategic interests of large firms promoting open weight and open source AI. To move forward, we need to build broader public AI ecosystems that ensure open source AI is accessible, trustworthy, and competitive with closed source alternatives.

## 2. Background on open source AI

Here, we briefly summarize open source software in ML and the role of open source AI projects in proving the viability of openness. This is not meant to comprehensively cover all the open software that plays a role in the ML research stack nor cover all successful open source AI projects, of which there have been many.

**Open Source in ML:** The machine learning community has embraced open source as both a cultural and technical norm. Software libraries like PyTorch (Paszke et al., 2019), Hugging Face Transformers (Wolf et al., 2019), and Diffusers (von Platen et al., 2022) have made advanced ML tools widely accessible. Open source codebases like

---

[1]Public AI Network [2]Current AI [3]Simon Fraser University [4]Kenyon College [5]AI Sweden [6]Metagov [7]University of Turku, Department of Computing [8]LAION [9]Juelich Supercomputing Center (JSC), Research Center Juelich (FZJ). Correspondence to: Joshua Tan <josh@publicai.co>, Nicholas Vincent <nvincent@sfu.ca>.

*Proceedings of the 43$^{rd}$ International Conference on Machine Learning*, Seoul, South Korea. PMLR 306, 2026. Copyright 2026 by the author(s).

OpenCLIP (Ilharco et al., 2021) and Megatron-LM (Shoeybi et al., 2019) have made training on supercomputers efficient and large-scale experiments reproducible. Evaluation suites like EleutherAI's `lm-eval-harness` (Gao et al., 2024) and LAION's CLIP Benchmark (Cherti & Beaumont, 2022) have become de-facto industry standards. Researchers routinely release code and models alongside publications, and open pre-trained weights have accelerated innovation and experimentation. Many foundational libraries like NumPy have long been released as open source software, which helped reduce reliance on proprietary scientific software. Relatedly, peer production platforms like Wikimedia projects (including Wikipedia) have long been central to ML training data (Johnson et al., 2024).

**Open Source AI Projects:** Various open source community communities have been extremely successful in producing AI projects, covering general models, domain specific models, datasets, benchmarks, and more. EleutherAI's Pile, GPT-NeoX and later Pythia models (Gao et al., 2020; Black et al., 2022; Biderman et al., 2023) provide fully open data and GPT models that have been widely used for practical and scientific purposes in language modeling. On the multi-modal front, LAION-400M / 5B image-text, Re-LAION-5B (Schuhmann et al., 2022; LAION, 2024), DataComp (Gadre et al., 2023), and LAION-audio-630k (Wu et al., 2022) projects were critical to providing open data for state-of-the-art vision and audio modeling. This enabled an open end-to-end pipeline for model training, resulting in fully open-source models like openCLIP (Cherti et al., 2023), LAION-CLAP (Wu et al., 2022) and openMaM-MUT (Nezhurina et al., 2025), as well as fully open-source text-to-image generative models such as Latent and Stable Diffusion (Rombach et al., 2022). Of special note is that at the time of their release, these community-built models matched or outperformed similar models from closed frontier labs. A wide array of other projects, including RWKV (Peng et al., 2023), BigScience (BigScience Workshop et al., 2023), BigCode (Li et al., 2023), OpenFold (Ahdritz et al., 2024), RedPajama and RedPajamas-INCITE (Weber et al., 2024), OLMo (Groeneveld et al., 2024), DCLM (Li et al., 2024), Pixmo & Molmo (Deitke et al., 2025), and more recently Marin (Hall et al., 2025) have all contributed to an ecosystem of usable open source AI.

## 3. Challenges for open source AI

Traditional open source software operates on the assumption that the full contribution cycle of use, modification, and redistribution is broadly accessible. While participation has never been perfectly equal, competent contributors can typically engage using commodity hardware and publicly available tooling. For large-scale AI models, this assumption breaks down.[1] Although smaller open models can be run locally, meaningful participation in model development requires capital, compute, and data infrastructure that most potential contributors lack.

### 3.1. Resource challenges

**Pretraining Requires Capital and Scale:** Modern models are trained on thousands of GPUs over weeks or months (Maslej et al., 2025) and require substantial energy (Luccioni et al., 2024). This demands not only access to ever-larger compute clusters–often only available to well-funded corporations or state-supported institutions–but also web-scale datasets, robust engineering teams handling super-computers, and complex distributed training infrastructure. Data collection is itself an extremely challenging process; merely downloading and storing large data sets (especially for multimodal training) presents a challenge for community contributors using commodity hardware.

**Post-training Depends on Private Data and Feedback Loops:** The fine-tuning, alignment, tool integration, and prompt orchestration that make models actually useful in practice are often kept closed. While model weights may be public, the systems that give them utility are private. Further, RLHF data generated by usage is typically siloed within individual platforms and not shared with the community, creating a compounding advantage for closed source and closed labs. Even when available, such RLHF and usage data require dedicated teams and additional resources to operationalize.

**Inference Isn't Cheap**: Unlike a software library which have comparatively trivial hosting costs, inference at scale (especially for large models) demands ongoing GPU access, orchestration systems, and cost management. And like RLHF, the usage data that flows from inference is hard to aggregate or share across organizations, limiting the ability of open models to improve from failures.

---

[1]In practice, frontier models require private complements such as compute and energy to be useful. In economic terms, that means that they are "impure public goods" (Reiss, 2021; Mazzucato et al., 2024) or club goods (Gries & Naudé, 2022) rather than pure public goods. A classic example of an impure public good is a lighthouse financed by "light dues" (Reiss, 2021) imposed on ship owners. The eventual economic model for AI could end up looking like this, though the question of who collects the dues will be salient to whether access to AI is democratized. For an example that more closely captures AI models, we might imagine a library whose collection is non-rival and openly licensed. As the catalog grows by orders of magnitude, it becomes impossible for typical users to find a book without hiring a private "guide" to help. The books remain true public goods, but access to their knowledge is mediated by a search-and-retrieval toll good, so access becomes "club-like" and effectively the library provides information as an impure public good.

## 3.2. Licensing challenges

While the costs of training, running, and maintaining a frontier model vastly exceed those of compiling and distributing traditional open source software, the challenges that the open source AI community faces go far beyond resource constraints.

**Sharing Data Is Risky and High-Friction:** Unclear copyright status, licensing complexities, and jurisdictional uncertainty create significant risk around data reuse, making it hard to share data within the open source community. For labs that train open-source models, it is often difficult to efficiently collect and deploy data from their deployers and inference providers—when there is any agreement at all to share such data.

**Licensing is Ambiguous or Fragile:** The layperson's understanding of "open source" AI is in fact more accurately termed open-weight AI. However, the term "open" in open-weight AI is misleading in two respects. For example, LLaMA's license contains restrictive terms and explicit revocability which have been widely criticized by the OSS community for being incompatible with established standards for open source (OSI Opinion & Maris, 2025). These concerns are not isolated, but extend across the industry as a whole. Companies like Meta can stop releasing models or add greater restrictions to future licenses at any time. Terms of service from other major providers impose additional constraints. For example, OpenAI's terms prohibit using outputs to develop models that compete with OpenAI (OpenAI, 2025). Anthropic's usage policy restricts how developer tools like Claude Code can be employed in third-party contexts. Restrictions like these, while commercially understandable, can limit independent evaluation, red-teaming, and reproducibility work upon which the research community depends.

**Transparency is Partial and Inconsistent:** More fundamentally, however, releasing weights alone does not provide the transparency that makes open source software auditable and reproducible. With traditional open source, access to the code brings with it the ability to understand how the software works along with the ability to reproduce, modify, and verify any claims about it. With open weights, by contrast, researchers often lack access to training data along with data curation decisions, RLHF procedures, and compute configurations. The "source" in open source once meant access to the complete blueprint, but open weights provide only the finished artifact.

While nonprofit and academic models maintain a commitment to openness, most powerful "open" models are trained by private companies and largely fail to provide details on training data or evaluation procedures. Opacity undermines the core benefits that openness typically provides. Without access to training data and post-training methods, external researchers cannot verify safety claims or fully understand reasons behind model behavior. They can only observe outputs as a form of phenomenological auditing that falls far short of the transparency that makes traditional open source software trustworthy. Attempts at independent evaluation face many structural barriers, and recent work on frontier AI auditing has documented the extent to which current arrangements leave companies "writing their own rules" (Brundage et al., 2026). When labs offer API access for external evaluation, sometimes accompanied by credits worth $1,000 or more, as is encouraged by the EU AI Act's Code of Practice, auditors may still face usage monitoring and access that can be revoked. Researchers often cannot verify which model version they are testing, whether it matches production systems, or whether safety behaviors differ between evaluation and deployment contexts.

## 3.3. Governance challenges

**Open Does Not Mean Safe:** A counterintuitive challenge for open source AI is that openly released models are often less safe than their closed counterparts. Open models are frequently research artifacts rather than deployment-ready systems. Safety work, including red-teaming, alignment tuning, and monitoring for emergent harms, requires sustained investment that volunteer communities and underfunded organizations cannot reliably provide. Furthermore, critical safety decisions occur during pre-training and data curation, not only during RLHF. By the time weights are released, significant portions of the safety-relevant design space have already been fixed.

**Closed-source Co-optation:** Open source is built on a compromise between community and commerce, but community-contributed evaluations, tooling, datasets, and fine-tuning techniques often accrue value to large firms whose commitment to open source is tenuous at best—in this case the frontier labs who train closed source models. Open source contributors imagine they are building shared infrastructure; in reality, they may be fueling a pipeline that concentrates power (Widder et al., 2024). Of particular concern to both open source *software* projects and the broader sphere of open or semi-open knowledge efforts like Wikipedia and StackOverflow is a trend whereby contributions that previously flowed into a commons pool instead flow into private ownership and begin to support a subsidized club good that looks like a public good.

Take the example of coding agents. With the release of more capable models in late 2025 (Gemini 3, Opus 4.5, Codex 5.2), software developers have increasingly adopted LLM coding agents like Claude Code into their workflows (Menlo Ventures, 2025). For small open source projects, LLM coding agents can substantially accelerate the process

of software development (Lewis, 2025): not just by assisting with the direct code writing process, but also with the knowledge work around planning, design and reproducing research code (Hua et al., 2025); or even reproducing data analyses required for investigative journalism (Hagar, 2026). With collaborative tasks too, agents are serving as a first-pass quality gate for issue triage (Wang et al., 2024; Feiglin & Dar, 2026) or reviewing merge requests (Cihan et al., 2025).

This can seem like a net positive for open source developers who want to contribute to a public good—however, access to the top-performing coding agents is gated through various means. Procuring substantial access to each coding agent often requires paying a hefty monthly subscription to each company, *and* it must be performed inside the company's proprietary model harness (e.g. Claude Code) that captures the user's exact process of software development. Open source model harnesses like OpenCode or OpenHands may provide open alternatives, but then access to the best models must be driven through API access, becoming several times more expensive than the direct subscription options.

Since it is most economically viable for a user to utilize a company's cheaper subscription offerings through a proprietary model harness, many open source developers are vulnerable to having their knowledge work be captured and transformed into implicit data labour (Vincent, 2026). Their user-agent interactions—prompting, responding to the model's questions, providing detailed feedback, providing source code snippets alongside error messages—and the cycles of iterative refinement which help produce the best quality work (Pan et al., 2025; Dai et al., 2025), also produce highly valuable annotations: annotations that include their preferences, acceptance criteria, problem-solving strategies, ideation processes, research and experimentation procedures, debugging strategies, potentially sensitive API keys, and sensitive documents on their filesystem—all directly captured by private model harnesses, and without clear governance mechanisms for requesting a data deletion.

**Access Without Accountability:** Proponents of the status quo suggest that AI access is increasingly free of charge. However, as social media has demonstrated, when users do not pay, they are often the product. Interactions can flow into the training pipelines with limited transparency about collection, retention, or use. Users may have little insight into how their data shapes future model behavior, limited ability to opt out of specific uses, and few mechanisms for recourse when their contributions are monetized. In this sense, users may still be paying for AI access, but with their data rather than their dollars.

**Expanding Gaps:** The AI ecosystem is evolving quickly. What began as token prediction from weights that fit on a local GPU is morphing into AI assistants that blend multi-modal reasoning, access to proprietary tools, and complex orchestration layers. As system complexity grows, the gap between "available weights" and "usable systems" widens. While there has been massive progress in terms of what can be run locally on consumer hardware (Gerganov, 2023; Ollama Team, 2025), these models still lack post-training alignment, retrieval augmentation, tool use integration, usage analytics, uptime guarantees, and continual updates that distinguish private, hosted services from merely downloadable weights.

Collectively, these challenges suggest that open source AI faces not a failure of values but of structure. The open source model, which flourished in an earlier era of low-cost computation and interoperable standards, is no longer sufficient on its own. To deliver on the promise of accessible and democratic AI (Collective Intelligence Project, 2024), we must build new public AI infrastructures that can provision and govern the full model lifecycle beyond just the first training checkpoint.

## 4. Position Statement

We assert: *open source AI, as currently practiced, will not by itself democratize access to AI or provision public goods as comparable open source efforts have done in other software categories.* Instead, we propose that open source AI must be embedded within a broader vision of public AI, defined by the following principles:

- Public Support: There must be public funding and infrastructure for inference, deployment, post-training, and data, beyond important pretraining.

- Public Access: Everyone—-researchers lacking resources, civic technologists, local communities outside of Big Tech—-must be enabled to build, adapt, and use competitive models.

- Public Accountability: Institutions accountable to the public—-governments, national labs, public utilities, universities, and nonprofits—must provision, host, and maintain models and related infrastructure.

- Private Commitments: Private actors must be encouraged (or required) to make commitments around openness, safety, and community control.

Public AI understands AI as a form of public infrastructure—think highways, libraries, water, or electricity. It is closely related to other forms of digital public infrastructure (DPI) including existing public digital stacks for identity, payments, and data exchange (Institute for Innovation and Public Purpose, UCL, 2026; Sieker et al., 2025; Tarkowski & Sieker, 2026).

# 5. Examples of public AI

Public AI is not a theoretical aspiration. Around the world, countries are already experimenting with concrete strategies for building and deploying large-scale AI systems in the public interest. We cover some examples of ongoing public AI efforts below:

**Creating New Foundation Models:** Faced with limits of reproducibility of important foundation model types, various organizations turned to public infrastructure and funds to create open versions of closed foundation models. Important examples are: (i) work done by BigScience initiative (spearheaded by HuggingFace), which trained BLOOM to replicate GPT class models on public supercomputer Jean Zay (IDRIS, France); (ii) non-profit organization LAION, which composed datasets and trained language-vision open-CLIP and language-audio CLAP models on various public supercomputers like JUWELS Booster (JSC, Germany) or Leonardo (CINECA, Italy); (iii) EleutherAI in US which collected Pile and trained models like GPT-NeoX-20B and Pythia, teamed up with various organizations including LAION on US public funded supercomputer SUMMIT (Oak Ridge Lab). Those self-organized efforts by grassroot communities executed on public infrastructure and backed up by public funds triggered a huge wave of follow-up open-source work including Stable Diffusion, RedPajama, FineWeb, DataComp, DCLM and so on, showing the potential of proper usage of public resources.

Most successful LLM efforts have primarily focused on high-resource languages (especially English), motivating efforts to create open-source models targeting other languages. For example CroissantLLM (Faysse et al., 2025), FinGPT (Luukkonen et al., 2023), GPT-SW3 (Ekgren et al., 2024), Minerva (Orlando et al., 2024), NorMistral (Samuel et al., 2025) and Poro (Luukkonen et al., 2025) in Europe have been trained primarily using public resources via compute grants from EuroHPC and national HPC organizations. Although these efforts succeeded in creating models with capabilities for specific languages, they also fragmented limited public resources to a point where no individual effort has had sufficient compute to create frontier models. Consequently, European efforts are increasingly focusing on creating broadly multilingual models such as EuroLLM (Martins et al., 2025), Salamandra (Gonzalez-Agirre et al., 2025), Teuken (Ali et al., 2025) and TildeOpen. The EU is also funding efforts to create open multilingual datasets (de Gibert et al., 2024) and models. The largest current project in the latter category is **OpenEuroLLM**, a consortium of 20 European institutions developing fully open foundation models. The project has so far released many smaller reference baseline models and was recently given access to EuroHPC strategic compute resources, providing it with compute totaling over 10M GPUh on four major

European HPC systems (Leonardo, LUMI, JUPITER and MareNostrum5). While the quality of these models are still unremarkable by the standards of model families like Qwen, DeepSeek, gpt-oss, Nemotron, or Kimi, they reflect a substantial investment in public AI in Europe that demonstrate the possibility of sustaining a genuinely open commons and democratic governance over how shared resources are used.

**Subsidizing Inference for Public Access:** Public infrastructure initiatives are emerging to address the computational barriers that limit access to advanced AI models. For example, the **Public AI Inference Utility**, a nonprofit organization, coordinates donated compute resources across multiple countries to provide free inference access for public and sovereign models (Public AI Inference Utility, 2025). As the primary deployer for models such as Switzerland's Apertus, it illustrates how distributed public infrastructure can sustain model accessibility beyond initial release. By pooling computational resources from diverse donors, the utility reduces the financial barriers to deploying and maintaining open models at scale. Initiatives such as the **National Deep Inference Fabric (NDIF)** provides researchers with shared access to open-weight models (Fiotto-Kaufman et al., 2025) and enables remote experimentation on model internals through standardized tools, addressing the "activation gap" between publicly available weights and usable research capabilities. This infrastructure allows researchers without substantial computational resources to conduct interpretability studies, fine-tuning experiments, and other investigations that require direct model access. These initiatives demonstrate complementary strategies for subsidizing inference both for broad public access and for specialized researchers.

**Auditing Standards and Benchmarks:** Public AI can mandate audit access as a condition of public funding, as well as maintaining versioned model checkpoints and separating infrastructure providers from the entities being evaluated. The recently launched AI Verification and Evaluation Research Institute (AVERI) has proposed a framework of "AI Assurance Levels" that illustrates a vision of public AI auditing infrastructure (Brundage et al., 2026). Level 1 is similar to current practice and involves limited third-party testing with constrained access. Level 4, on the other hand, would provide "treaty grade" assurance sufficient for international agreements on AI safety. Current private auditing arrangements largely operate at Level 1, but public AI infrastructure could enable Level 3-4 assurance as a baseline expectation with mandated access, versioned checkpoints, and separation between infrastructure providers and evaluated entities. Public AI initiatives are also developing comprehensive evaluation frameworks that address multilingual and multicultural capabilities. For example, **SEA-HELM** (Southeast Asian Holistic Evaluation of Language Models), developed in collaboration with AI Singapore, provides a rigorous evaluation suite emphasizing Southeast Asian languages across

five core pillars: NLP Classics, LLM-specifics, SEA Linguistics, SEA Culture, and Safety (Center for Research on Foundation Models, 2025). Supporting Filipino, Indonesian, Tamil, Thai, and Vietnamese, SEA-HELM demonstrates how public AI infrastructure can establish evaluation standards that go beyond English-centric benchmarks to ensure models serve diverse linguistic and cultural communities.

New approaches are still being developed, including the Airbus for AI (Valero & Crespo, 2024; Tan et al., 2025) and CERN for AI (CAIRNE; Juijn et al., 2024) proposals for multilateral visions of public AI.

## 6. Alternative Views

We recognize a number of serious alternative views that challenge the necessity or feasibility of public AI.

### 6.1. View 1: The Market Is Working. Let OpenAI and Meta Lead.

Many believe that the private sector is successfully scaling AI access. OpenAI, Meta, Mistral, and DeepSeek have made advanced models available cheaply or for free. Proprietary labs have shown tremendous speed and capability in model iteration, evaluation, and deployment. Their models are at the performance frontier, their user experience is polished, and their costs are rapidly dropping.

Response: Access is not governance, and it is not sovereignty. These systems remain opaque and subject to unilateral revocation. For example, it has been reported that LLaMA 4 will be the final model release in the LLaMA family, with Meta shifting their focus towards closed weight models (Vaughan-Nichols, 2026). The ability to use a chatbot today also does not ensure access to trustworthy, auditable systems tomorrow. As another example, Alibaba Qwen removed access to the free version of Qwen Code in April 2026 (Lanz, 2026). Public AI is not about replacing private labs, but about ensuring that there are durable, open, and accountable systems aligned with public needs and values. For example, the USA's National Deep Inference Fabric was designed to provide democratic access to open-weight models (Fiotto-Kaufman et al., 2025), while Sweden's GPT-SW3 (Ekgren et al., 2024) was initially trained to address ChatGPT's poor performance in Swedish and other Scandinavian languages.

It is also worth noting that Meta and other corporate-OSS builders stand to benefit if open source inference becomes publicly funded: the cheaper the inference, the more value accrues to the application layer.

### 6.2. View 2: Open Source Will Win Eventually. Just Be Patient.

This view argues that the open source ecosystem is improving rapidly (Maslej et al., 2025) and will eventually produce models on par with or better than proprietary models. The release of high-quality weights (e.g., Mistral, DeepSeek, Zephyr), coupled with open fine-tuning libraries and model merging techniques, suggests that community-driven innovation will outcompete closed models in the long run.

Response: Open source progress has been remarkable. There have been many academic, nonprofit, and community-led efforts to train foundation models. However, most of the strongest and widely used open models today were pre-trained by well-capitalized private companies: compare LLaMA 3.1-8B's 6M monthly downloads on Hugging Face with EleutherAI Pythia's 900k and OLMo 3-7B's 170k (as of late Jan 2026) (Hugging Face, 2026). LLaMA is also broadly adopted in downstream models like VLMs (eg, LLaMA3-Nemotron, etc.), while the nonprofit versions lack comparable adoption as components. Notable exceptions that confirm the rule are models by LAION like language-audio CLAP (14M monthly downloads as of Jan 2026) and openCLIP variants (per model, between 1M-2M downloads per month, exceeding >60M all time downloads). LAION was backed up by public infrastructure like supercomputers and storage in its work, and evidence shows it is in such cases possible to be on par with private entities. Open options also do not compare well with the adoption seen by OpenAI and Anthropic—not to mention the potential for extraordinary adoption as proprietary models are incorporated into product platforms like Google Search or Microsoft Office. Open source may or may not be beat by closed source, but additional public investment is very unlikely to hurt and may prove critical to future sustainability and competitiveness. Public AI also ensures that open source models remain accessible, trustworthy, and responsive to broad public needs rather than to the incentives of a single commercial sponsor.

### 6.3. View 3: OSS + Hosting Already Works.

Why add bureaucracy? A practical open source ecosystem is already in place. Open models are hosted via Hugging Face, Replicate, and Open Router. Inference is affordable. User-facing products are emerging. Why burden this with new governance structures or public spending?

Response: Like view 1, this view confuses current availability with long-term stability. Most current deployments rely on ephemeral commercial hosting or terms that can be revoked. The fragility of the OSS+hosting stack is exemplified by the LLaMA license and the risk of unilateral pullback from companies like Meta. Public AI does not aim to replace this ecosystem but to underwrite it. This already happens,

especially for academic and nonprofit projects: for example, national labs in the US use EleutherAI's GPT-NeoX and have provided some support for the project, while the French National Center for Scientific Research supported the BigScience project, which trained BLOOM on Jean Zay, a French public supercomputer (BigScience Workshop et al., 2023). LAION's work on openCLIP (Schuhmann et al., 2022; Cherti et al., 2023; Nezhurina et al., 2025) was also enabled and supported by public compute and storage backed by the grants from the Gauss Center for Supercomputing at the Juelich Supercomputing Center, a public research facility in Germany hosting publicly funded supercomputers.

### 6.4. View 4: Regulation Is a Better Tool Than Public Investment.

Instead of building new infrastructure, governments can simply regulate AI development—imposing transparency requirements, safety standards, and licensing constraints. Regulatory frameworks such as the EU AI Act and export controls on GPU sales aim to shape the AI landscape through law rather than through investment.

Response: Regulation is essential, but it is not sufficient. It can curb harmful behavior but does little to guarantee access, usability, or equitable participation. Public AI is proactive: it builds capabilities and institutions that embody public values from the outset. Rather than rely solely on constraints imposed on private actors, public AI enables public-purpose development from the ground up. This complements regulation by demonstrating and institutionalizing best practices. For example, Canada's SCALE AI project funds both regulatory and capability-building efforts, providing shared infrastructure for data and training.

### 6.5. View 5: Public AI Will Be Inefficient and Capture-Prone.

There is a long history of inefficient or mismanaged public-sector technology projects. Bureaucracies move slowly, are vulnerable to capture, and can't attract talent (Mazzucato, 2013). Why should we expect public AI to be different?

Response: This is a valid concern. However, well-governed public institutions do exist and have produced extraordinary technological advances—from GPS to the internet to the Hubble Space Telescope. Our position is not that governments should necessarily divert funds from other priorities toward AI, but rather that the public money already being spent on AI (for example, on procurement of AI goods and services) should be structured to better serve the public interest. Moreover, public AI need not be synonymous with government-only models. Public funding could support existing nonprofit activities—even OpenAI once contemplated asking for public funding (Klein, 2021). Proposals like Air-

bus for AI (Tan et al., 2025) envision a hybrid, multilateral structure of many national entities, each organized as public utilities. Successful examples like ERC, CERN, and W3C show that public AI can be designed to resist capture and reward quality.

## 7. Technical and Societal Implications

Public AI shifts the focus of machine learning research away from monolithic frontier labs and toward shared infrastructure, cooperative development, and inclusive deployment. For the ML community, this has far-reaching implications:

- For ML Researchers: Shared model libraries and pooled inference capacity democratize frontier experimentation. When models are shared, researchers can access and intervene on the internals of LLMs and other models without the cost or complexity of hosting their own hardware (Bommasani et al., 2021; Fiotto-Kaufman et al., 2025). They can access more of the RLHF and query data that is essential to frontier model capability research. Public AI also reduces fragility and promotes reproducibility across labs.

- For Non-CS Fields: Domains like healthcare, education, and law increasingly require high-quality models. Public AI enables domain experts to adapt systems to local needs without relying on private APIs or black-box deployments.

- For Open Source Ecosystems: Many contributors now work without guarantees that their outputs will remain in the commons. Public AI ensures their efforts resist private capture and support genuinely open systems.

- For Governments and Funders: Public AI can serve as a key plank of digital sovereignty and national innovation strategies (Public AI Network, 2024). Governments can focus investment on shared infrastructure and safety rather than competing on consumer UX.

- For the Broader Public: Public AI supports democratic accountability and contestability. It embeds collective input in how powerful systems are developed and used.

## 8. Conclusion

The machine learning community should not conflate open source with public good. We argue for a future in which open source AI is nested within public AI infrastructures: institutions and commitments that activate, sustain, and distribute AI systems for the public benefit.

If the goal is to enable a diversity of actors to build and deploy capable models, then we must move beyond a romantic view of open source and begin investing in AI as public infrastructure. If open source is to win, it must go public.

## Acknowledgements

We would like to thank Stella Biderman, Nathan Lambert, Imanol Schlag, and many others for helpful comments in the writing of this paper.

JJ acknowledges funding by EU Horizon under grant no. 101214398 (ELLIOT) and co-funding by EU from Digital Europe Programme under grant no. 101195233 (openEuroLLM), co-funding from EU under Digital Europe Programme under grant no. 101198470 (LLMs4EU), from EuroHPC Joint Undertaking Programme under grant no. 101182737 (MINERVA), and funding by the German Federal Ministry of Research, Technology and Space (BMFTR) under the grant 16HPC117K (MINERVA).

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

## A. Comparing select open source software and open models

| Properties | Linux | scikit-learn | TensorFlow | Kubernetes | OLMo | DeepSeek | LLaMA |
|---|---|---|---|---|---|---|---|
| Type | Operating System | ML Library | ML Framework | Container Orchestration | AI Model | AI Model | AI Model |
| Transparency | High — development is fully visible | High — all algorithms and tests are publicly documented and peer-reviewed | Medium — public codebase, but production usage depends on internal forks | High — development processes, governance, and roadmap are fully public | High — training pipeline, data decisions, and documentation openly shared | Medium — some training details and weights released, but pipeline unclear | Low — no access to training data, limited documentation, opaque post-training |
| Community Governance | Yes — community + Linux Foundation | Yes — consensus driven; backed by research orgs | Yes — SIGs, GitHub issues, TF RFCs | Yes — CNCF technical governance | Yes — AI2 hosts calls, roadmap, accepts contributions | No — releases set by DeepSeek | No — decisions made by Meta; no public forum |
| License Stability | Clear | Clear | Clear | Clear | Clear | Unclear | Unclear |
| Use Without Large Infra | Yes; runs on typical personal hardware | Yes; any Python env | Yes; CPUs or GPUs; many hosted options | Yes; single-node or small clusters | Partially; some GPUs locally; pruning supported | No; inference targets powerful clusters | No; needs high-end GPUs |
| Open Source Maintenance | Active; broad community + LF | Active; INRIA-led core + community | Active; Google + community | Active; CNCF + industry | Active; AI2 with public roadmap | Partial; periodic checkpoints | Irregular; Meta-driven |
| Business Model | Service-based (Red Hat, etc.), donations | Academic; grants/volunteers | Freemium support by Google | Cloud-vendor support via CNCF | Non-profit; philanthropic | Hedge-fund backed; opaque | Meta strategic positioning |
| Supporters/Adopters | Universities, enterprises, hobbyists, clouds | Universities, educators, research | Enterprises, researchers, hobbyists | Global enterprises, cloud providers | Academic labs, open-science advocates | Emerging China-centric dev community | Academic labs, startups via HF |

*Table 1.* Comparing open source software and open source AI projects along a variety of axes, including transparency, governance, licensing, and maintenance concerns. A key takeaway is that AI is not like other open source software.

