# OpenReview forum: "Position: If Open Source Is to Win, It Must Go Public"
_ICML.cc/2026/Position_Paper_Track — ICML 2026 Position Paper Track spotlight_

### Official Review · Reviewer_DYow · 2026-03-12

**Significance:** 4
**Argument Clarity:** 4
**Rating:** 5
**Confidence:** 4

**Questions:**

No question at this point

**Alternative Views Section:**

Yes

**Compliance With Llm Reviewing Policy A Conservative:**

Affirmed.

**Discussion Potential:**

4

**Final Justification:**

The paper and the advocated position is very clear and well discussed. The rebuttal addressed my concerns. I am keeping my positive score.

**Paper Summary:**

This paper questions the  role of public institutions and infrastructures in a context of promoting  open-source AI. The position is that their roles should be emphasized, and the core debate revolves around innovation, sovereignty, safety, and market structure. After recalling notable open source and open weights initiative in ML/AI, in a broad but coherent overview, authors discuss challenges associated to ressources, licensing and governance. After clearly stating thgeir position, they giove examples of such public initiatives that support their views. Alternative views (e.g. why changing the current situation, which seems to work ? should regulation be prioritized ?) are also clearly discussed,

**Position:**

Yes

**Position In Title:**

Yes

**Related Work:**

4

**Strengths And Weaknesses:**

The paper and the advocated position is very clear and well discussed. It is supported by numerous examples and evidences. The topic is deemed very relevant to the ICML community by my reviewer's opinion, and is likely to inspire interesting discussions. I strongly advocate for its acceptance in ICML program.

On maybe a more personal note/view, I find that though listed in the challenges part of this paper, the questions on how and why governments and public insitutions should prioritize fundings on AI related open source models could be more discussed. For example, this public money could be better invested on other important topics (healthcare, mitigating effects on climate changes, public education, fightinh against poverty, etc.) Safety concerns, regarding how well safeguards can be preserved while maintaining open source policies on how code / weights are released, might also be a little bit more discussed, or be considered as an alternative view.

**Support:**

4

---

> ### Author Rebuttal · Authors · 2026-03-30
>
> Thank you for your thoughtful and supportive review, we appreciate the constructive suggestions and address them below:
>
> On the opportunity cost of directing public funding towards AI: This is an important point, but we would note that governments are already spending heavily on AI. For example, Japan recently announced $6 billion in support for domestic AI development, channeled primarily through a consortium of private firms led by SoftBank. Our position is not that governments should divert funds from other priorities toward AI, but rather that the public money already being spent on AI should be structured to better serve the public interest. We will further clarify this key point.
>
> On safety concerns with open source models: We discuss this in Section 3.3 highlighting that "Open Does Not Mean Safe". Precisely because openly released models often lack sustained safety investment, public institutions are needed to back the deployment and maintenance of open models with the resources and accountability that volunteer communities cannot reliably provide.
>
> Thank you again for engaging deeply with the work.

---

> > ### Author Rebuttal · Reviewer_DYow · 2026-04-03
> >
> > Thank you for the rebuttal which I read carefully, which partially resolved my questions:
> >
> > Though I concur to some extent with this sentence
> > > we would note that governments are already spending heavily on AI.
> >
> > I still believe that those choices are debatable and could serve as a counter-argument to developing public opensource AI.
> >
> > I nevertheless keep my score as I believe that this paper is a good fit for the position paper track.

---

### Official Review · Reviewer_aDog · 2026-03-12

**Significance:** 3
**Argument Clarity:** 3
**Rating:** 5
**Confidence:** 4

**Questions:**

- Is there any environmental concerns with making AI models more freely available?
- Who do you suggest should maintain and control these centers for AI?
- Any political concerns with AI infrastructure being funded by public institutions (governments)?

**Alternative Views Section:**

Yes

**Compliance With Llm Reviewing Policy A Conservative:**

Affirmed.

**Discussion Potential:**

4

**Final Justification:**

I have not changed my mind and am still positive about this paper's potential to lead to a good discussion.

**Paper Summary:**

The authors argue that public institutions should invest in AI infrastructure in order to ensure that these models are accessible broadly and able to compete with corporate-backed solutions.

**Position:**

Yes

**Position In Title:**

Yes

**Related Work:**

3

**Strengths And Weaknesses:**

## Strengths

- The paper is well-written and supports its arguments with extensive discussions and background.
- The topic is timely and important.
- Alternative views are reasonable and discussed thoroughly.

## Weaknesses

- The title is catchy but ambiguous. Please consider stating your position more clearly.
- The financing issues are not discussed much at all, even though this is of course the major difficulty in realizing this position. It would be good to dig deeper into the financial models adopted by the countries and institutions in section 5.
- There would be more potential for discussion if the specifics regarding financing were outlined more clearly. As it stands, it is hard to disagree that there should be more publicly available AI infrastructure.

**Support:**

3

---

> ### Author Rebuttal · Authors · 2026-03-30
>
> Thank you for the supportive review and the constructive feedback.
>
> On the title: We appreciate the note and will consider making the position more explicit.
>
> On financing: We acknowledge that business models and financial sustainability is an important consideration. However, we would note that substantial public financing of AI is already happening e.g. Japan's recent $6 billion commitment, EuroHPC compute grants, AI Factories, AI Gigafactories, the US NAIRR etc… without a clear path to profitability. The question is less whether governments will fund AI, and more how that funding is to be structured and distributed to serve public interests.
>
> On environmental concerns: With or without public AI, people are already consuming AI at massive scale. Our position is largely agnostic on this front. We are not arguing for making AI more freely available per se, but for public institutions to play a role in its infrastructure. That said, public involvement could plausibly improve efficiency: governments are more likely to invest in interoperability and resource-sharing across HPC systems, or to enforce usage of renewable energies to power the facilities, whereas private actors have little incentive to do so. Centralizing inference through coordinated public infrastructure could be more energy-efficient than the current landscape of fragmented private deployments. Further, one point we can add is that public AI actors have substantially stronger incentives to take actions like surfacing environmental costs in AI interfaces.
>
> On who should maintain these centers: As we discuss in Section 5, there are already many public AI initiatives underway across different countries and institutions. Our paper is less about proposing a new centralized body and more about providing a unifying philosophy that encourages these existing efforts to collaborate around shared goals of public access, accountability, and sustainability.
>
> On political concerns: These are real, and we discuss the risk of capture and inefficiency in Section 6.5. We are not advocating for AI to be exclusively governed by public institutions, but rather for more balance in a status quo where AI development and deployment is almost entirely controlled by private organizations. Public AI need not mean government-run AI; successful models like CERN, ERC, and W3C show that publicly funded institutions can resist political capture while maintaining accountability.
>
> Thank you once again for your thoughtful critiques.

---

> > ### Author Rebuttal · Reviewer_aDog · 2026-04-01
> >
> > Thanks for the detailed rebuttal and for the clarifications! I have a couple of quick follow-ups.
> >
> > > The question is less whether governments will fund AI, and more how that funding is to be structured and distributed to serve public interests.
> >
> > I am not sure I completely agree and it would still improve the paper if you paid more attention to this topic in it.
> >
> > > On environmental concerns: With or without public AI, people are already consuming AI at massive scale. Our position is largely agnostic on this front.
> >
> > Your position is, as far as I understand it, that AI should be more widely available. How could that not increase usage and possibly lead to increased externalities?

---

### Official Review · Reviewer_498D · 2026-03-26

**Significance:** 3
**Argument Clarity:** 3
**Rating:** 5
**Confidence:** 3

**Questions:**

It would be great to mention a timeline/plan for people in public sector on what is required from them to enable these goals. Ideally, there should be a section that shows how this new position/paradigm can handle all the issues mentioned in section 3.

**Alternative Views Section:**

Yes

**Compliance With Llm Reviewing Policy A Conservative:**

Affirmed.

**Discussion Potential:**

4

**Paper Summary:**

The paper proposes a position to go beyond open-weight and open-source AI models to treating or supporting AI as a public infrastructure, just like roads, hospitals, or, more closely, digital payment systems. The authors present issues with the existing paradigm, where even though open-weights model are released in many cases, they are not open-sourced. Moreover, there is a barrier to entry for the public to evaluate and try these models due to the requirement of expensive computing. The authors also argue for this position against several strong alternative views, such as a free market hypothesis or the use of governmental regulations instead of AI as public goods.

**Position:**

Yes

**Position In Title:**

Yes

**Related Work:**

4

**Strengths And Weaknesses:**

# Strengths

* The paper is well-written, well-structured, and provides extensive evidence in support of the position it is trying to make. This is done by providing several limitations of the existing paradigms related to both computing and data, and challenges in evaluating closed-source models. I particularly like the comparison to other public goods/services.
* The authors provide several strong alternative views and also discuss their limitations well in accordance with their positions. This is likely to create good discussion potential in the community.
* The paper also provides good examples of programs already taking place in several countries in alignment with this position.

# Weaknesses
* While the authors present the limitations of the existing paradigms well, they should also address how a public system would be able to address those limitations and would not fall into other issues due to added bureaucracy. For example, the authors mention that Licencing is ambiguous and fragile in the existing paradigm. But it is unclear how a public system would resolve that or would even be able to obtain data at scale to be public?
* Similarly, the authors should mention potential plans for how governments/people working in the public sector could enable such projects. While the goals might be noble and good, without concrete plans in support of this position, it would be challenging for a reader to understand how these things can be implemented.

**Support:**

2

---

> ### Author Rebuttal · Authors · 2026-03-30
>
> Thank you for the thoughtful review and for engaging closely with the paper.
>
> On the weaknesses raised: We appreciate these concerns and want to clarify that our paper does not claim to have all the solutions to the challenges we identify. Rather, as a position paper, our goal is to provide a unifying philosophy that encourages the many public AI initiatives already underway (as discussed in Section 5) to properly identify what are the most pressing deficits and needs that have to be covered by public AI (eg compute, independent validation, reproducibility, etc) and to collaborate along those lines more effectively around shared goals of access, accountability, and sustainability. We believe articulating this shared vision is a necessary first step before concrete implementation plans can converge.
>
> On licensing and data challenges specifically: we do not claim that public institutions will automatically resolve these issues. But we would like to note that many of the successful open source AI efforts we discuss (BLOOM, LAION/openCLIP, Pythia) were enabled precisely by public infrastructure and funding, and managed to navigate these challenges in practice, which also helped the community to see how licensing can be addressed. Public institutions also have tools that private actors lack, such as the ability to set data-sharing conditions on publicly funded research.
>
> On the risk of added bureaucracy: this is a valid concern, and we discuss it in Section 6.5. We acknowledge that public institutions can be inefficient and capture-prone, but successful models like CERN, ERC, and W3C demonstrate that well-designed publicly funded institutions can produce extraordinary results. So this becomes a question of proper implementation of public governing that avoids burdens of bureaucracy, rather than being a fundamental limitation that cannot be worked around. The greater risk, in our view, is the absence of any meaningful public alternative to concentrated private control over AI infrastructure.
>
> Thank you once again for the clarifications raised.

---

> > ### Author Rebuttal · Reviewer_498D · 2026-04-03
> >
> > The authors have fairly addressed the pointed weaknesses. I agree that a position paper should not have all the solutions, but at least some directions that can initiate a conversation between the interested parties. For example, as the rebuttal mentions successful cases of super large-scale public science projects like CERN, ERC, etc. Maybe the authors can add this to the paper, with a historical analogy of how going public for solved problems with particle physics, etc., at the time, and how similar directions can help with AI.
> >
> > I think this might be a relatively larger update, but it would greatly help the reader's understanding of the reader towards the position of this paper. I'm keeping my rating at an accept.

---

### Decision · Program_Chairs · 2026-04-30

**Decision:**

Accept (spotlight)

**Comment:**

The paper addresses an important limitation of existing open source paradigms and argues for an alternative where public resources make open source truly accessible. Reviewers appreciated the comparison to other public goods/services, strong alternative views section, and compelling evidence in support of the central argument (including examples of programs in some countries).